# Development of Injectable PEGylated Liposome Encapsulating Disulfiram for Colorectal Cancer Treatment

**DOI:** 10.3390/pharmaceutics11110610

**Published:** 2019-11-14

**Authors:** Mohammad Najlah, Ammar Said Suliman, Ibrahim Tolaymat, Sathishkumar Kurusamy, Vinodh Kannappan, Abdelbary M. A. Elhissi, Weiguang Wang

**Affiliations:** 1Pharmaceutical Research Group, School of Allied Health, Faculty of Health, Education, Medicine and Social Care, Anglia Ruskin University, Bishops Hall Lane, Chelmsford CM1 1SQ, UK; ammar.said-suliman@anglia.ac.uk (A.S.S.); ibrahim.tolaymat@anglia.ac.uk (I.T.); 2Faculty of Science & Engineering, University of Wolverhampton, Wolverhampton WV1 1LY, UK; S.Kurusamy2@wlv.ac.uk (S.K.); v.kannappans@wlv.ac.uk (V.K.); w.wang2@wlv.ac.uk (W.W.); 3College of Pharmacy and Office of the Vice President (Research and Graduate Studies), Qatar University, Doha, Qatar; aelhissi@qu.edu.qa

**Keywords:** disulfiram, copper, liposomes, colorectal cancer, chemoresistance, PEGylation

## Abstract

Disulfiram (DS), an anti-alcoholism medicine, shows strong anti-cancer activity in the laboratory, but the application in clinics for anti-cancer therapy has been limited by its prompt metabolism. Conventional liposomes have shown limited ability to protect DS. Therefore, the aim of this study is to develop PEGylated liposomes of DS for enhanced bio-stability and prolonged circulation. PEGylated liposomes were prepared using ethanol-based proliposome methods. Various ratios of phospholipids, namely: hydrogenated soya phosphatidylcholine (HSPC) or dipalmitoyl phosphatidylcholine (DPPC) and *N*-(Carbonyl-methoxypolyethylenglycol-2000)-1,2-distearoyl-sn-glycero-3-phosphoethanolamine (DSPE-PEG_2000_) with cholesterol were used. DS was dissolved in the alcoholic solution in different lipid mol% ratios. The size of the resulting multilamellar liposomes was reduced by high-pressure homogenization. Liposomal formulations were characterized by size analysis, zeta potential, drug loading efficiency and stability in horse serum. Small unilamellar vesicles (SUVs; nanoliposomes) were generated with a size of approximately 80 to 120 nm with a polydispersity index (PDI) in the range of 0.1 to 0.3. Zeta potential values of all vesicles were negative, and the negative surface charge intensity tended to increase by PEGylation. PEGylated liposomes had a smaller size (80–90 nm) and a significantly lower PDI. All liposomes showed similar loading efficiencies regardless of lipid type (HSPC or DPPC) or PEGylations. PEGylated liposomes provided the highest drug biostability amongst all formulations in horse serum. PEGylated DPPC liposomes had t_1/2_ =77.3 ± 9.6 min compared to 9.7 ± 2.3 min for free DS. In vitro cytotoxicity on wild type and resistant colorectal cancer cell lines was evaluated by MTT assay. All liposomal formulations of DS were cytotoxic to both the wild type and resistant colorectal cancer cell lines and were able to reverse chemoresistance at low nanomolar concentrations. In conclusion, PEGylated liposomes have a greater potential to be used as an anticancer carrier for disulfiram.

## 1. Introduction

The medical need for better cancer therapies is undiminished, while drug development is slow and costly, mainly due to the large risk of toxicity of novel molecules. Development of a new drug takes, on average, 15 years and costs US$1.5bn, with only 5–25% of new oncology drugs in clinical development actually reaching the market [1]. This dilemma has led to a booming interest in repurposing of known drugs into new use in Europe and the USA [2]. Currently, over 30,000 drugs have been on the market. Considering their derivatives, this is a tremendously huge resource for drug repositioning.

Disulfiram (DS, Figure 1), a well-known anti-alcoholism drug that has been used safely for over 65 years, has shown potent anticancer activity against the aggressive form of colon, breast, lung, prostate, ovarian, cervical and brain cancers. Furthermore, this drug specifically and effectively terminates drug-resistant cancer stem cells (CSCs) and reverse chemoresistance [3,4]. DS also has a significant synergic cytotoxicity with a wide range of first-line anticancer drugs such as cisplatin, 5-flurouracil, paclitaxel, gemcitabine, doxorubicin and temozolomide in vitro and saves normal cells in kidney, gut and bone marrow *in vivo* by increasing the therapeutic index [5,6,7]. The anticancer activity of DS is copper (II)-dependent as DS strongly chelates Cu to form a DS/Cu complex [8] (Figure 1). Cancer tissues possess significantly higher copper levels than their normal counterparts [9]; this may grant DS the selectivity to target cancer cells [10]. Although DS shows strong anticancer activity in laboratory, its application in cancer clinics is highly limited by its bio-instability. The half-life of DS in the blood stream is less than 4 min [11,12].

The rapid degradation is the major challenge faced when using DS in cancer therapy. Our work demonstrated that the intact thiol groups in DDC are essential for chelating Cu2 and targeting cancer [13]. Currently, only an oral version of DS is available in clinics. The thiol groups are instantly methylated or glucuronidated in the liver. While the DS metabolites are active against alcoholism, the anti-cancer activity does require unmodified DS [13]. The development of an efficient delivery system able to protect DS during its circulation into cancerous cells is essential. There has been a growing interest in developing nano-drug-delivery systems that are able to provide sufficient protection to DS, thus, enabling clinical trials [14].

Liposomes are nanocarrier systems made of relatively stable and cheap compositions and prepared by hydration of phospholipids known to be non-toxic, biodegradable, biocompatible, and non-immunogenic [15,16]. The unique structures of liposomes make them capable of encapsulating both hydrophilic and lipophilic drugs [17,18]. For example, Wang et al. have successfully encapsulated DS into liposome (Lipo-DS) and managed to mildly extend the half-life of DS in the bloodstream to approximately 20 min [3,19]. However, more development is required in this field, especially for enhancing the loading efficiency and extending the stability of DS [14]. For example, developing long-circulating drug delivery systems that might have a potential for translation to cancer therapy. Furthermore, the instability of liposomes owing to liability of its compositions to hydrolysis, oxidation and microbiological contamination needs to be overcome.

One approach to preparing liposomes is by using the proliposome technology as stable precursors that can instantly generate vesicles prior to formulation administration [20]. Ethanol-based proliposomes are based on using ethanolic lipid solutions to facilitate the hydration of phospholipids and generate oligolamellar or multilamellar liposome vesicles [21]. Proliposomes represent an economical approach to scale-up liposome production and resolve the instability problems associated with conventional thin-film-made liposomes. These advantages are attributed to the predominance of the ethanolic phase in proliposomes instead of water, avoiding phospholipid hydrolysis and offering self-antimicrobial preservation [22,23].

In this study, we have developed injectable DS-loaded PEGylated liposomes for colorectal cancer treatment. Ethanol-based proliposome technology was used to prepare DS-loaded liposomes with various lipid compositions. The prepared formulations were characterized by particle size; particle size distribution; zeta potential and drug entrapment efficiency. Additionally, the cytotoxicity of the PEGylated liposome encapsulated DS was examined in both 5-fluorouracil sensitive (H630_WT_) and resistant (H630_R10_) colon cancer cell lines.

## 2. Materials and Methods

### 2.1. Materials

Hydrogenated phosphatidylcholine (HSPC; Phospholipon 90H), dipalmitoyl phosphatidylcholine (DPPC) and *N*-(Carbonyl-methoxypolyethylenglycol-2000)-1,2-distearoyl-sn-glycero-3-phosphoethanolamine, (DSPE-PEG_2000_) were obtained from Lipoid, Steinhausen, Switzerland. Tetraethylthiuram disulfide or disulfiram (DS) (97% pure) and Tween^®^ 80 were purchased from Acros Organics, Loughborough, UK. Dulbecco’s modified Eagle’s medium (DMEM), non-essential amino acid solution and *L*-glutamine (cell culture tested, 99.0–101.0%), Trypsin-EDTA solution, ethanol (absolute and 70%), 96-well plates (sterile with lids), 50 mL centrifuge tubes (sterile), tissue culture flask 75 cm^2^ (sterile) and serological pipettes (sterile) were obtained from Fisher Scientific, Loughborough, UK. Cholesterol (Ch; ≥ 99%), glass vials (15 mL), fluorouracil (5FU), dimethyl sulfoxide (DMSO), thiazolyl blue tetrazolium bromide, fetal bovine serum (FBS), phosphate-buffered saline (PBS) tablets, trypan blue solution (0.4% liquid, sterile filtered), syringe filters (0.2 and 0.45 µm), syringe needles and sterile pipette tip boxes were purchased from Sigma Aldrich, Dorset, UK. Horse Serum, New Zealand origin was purchased from Gibco, Fisher Scientific, Loughborough, UK. Colorectal cell line (H630_WT_) and acquired resistance to 10 µM 5FU colorectal cell line (H630_R10_) were obtained from Professor Weiguang Wang Group, University of Wolverhampton, Wolverhampton, UK. All other reagents were of pharmaceutical grade and used as received.

### 2.2. Methods

#### 2.2.1. Preparation of Liposomes

Liposomal formulations were prepared using the ethanol-based proliposome method reported previously [20]. The lipid phase (phospholipid: Ch; 1:1 mole ratio) (300 mg) was dissolved in absolute ethanol (300 μL) at 70 °C (water bath) for 1 min within a 30 mL glass vial. For the PEGylated liposomes, the ratio was DSPE-PEG_2000_: phospholipid: Ch, 0.1:0.9:1 mole ratio. Disulfiram (DS) was then added in the ethanolic solution to produce a range of concentrations as mol% of the ultimate lipid phase (0, 5, 10 and 15 mol% of the lipid phase, Table 1). Aqueous (water) phase (30 mL), heated significantly above the Tm of the lipid, was added immediately to avoid lipid phase solidification. Liposomes were generated upon vigorous hand shaking and vortex mixing (Grant-bio PV-1, Shepreth, UK) for 5 min. Liposomal formulations were then kept for annealing above the phase transition temperature of the lipids for 2 h followed by size reduction.

#### 2.2.2. Size Reduction of Liposomes

Liposome dispersions (30 mL, 10 mg formulation/mL) was placed in the bench top NanoDebee high-pressure homogenizer (Bee International Inc., Northampton, UK) and processed for 10 cycles through seven reactors at 20,000 psi. Thereafter, liposome dispersions were placed on a magnetic stirring plate at 350 rpm in a fume-hood to stir for 5 h to minimize the trace amounts of ethanol in the formulation.

#### 2.2.3. Photon Correlation Spectroscopy (PCS) Analysis after Size Reduction

The photon correlation spectroscopy (PCS) technique relies on the Brownian motion of the particles using the Zetasizer instrument (Zetasizer nano, Malvern Instruments Ltd., Malvern, UK). The size and polydispersity of the homogenized liposomes were analyzed by recording the hydrodynamic diameter (Z_average_) and polydispersity index (PI), respectively, using the Zetasizer instrument (Zetasizer nano, Malvern Instruments Ltd., Malvern, UK).

#### 2.2.4. Determination of Drug Entrapment Efficiency

Entrapment efficiency of DS was determined by adapting the separation methods previously reported [24,25]. One mL of liposome suspension was passed through a syringe filter (Durapore^®^ Membrane PVDF Filters, HVLP02500, 0.45 µm, Hertfordshire, UK) followed by injecting 2 mL distilled water to wash the filter. The filtrate (0.5 mL) was add to 1.5 mL of methanol. The mixture was bath sonicated (Fisherbrand™ 112201, Loughborough, UK) for 2 min to disrupt the liposomes and release the entrapped drug. To determine the total amount of DS in liposome formulation, 1 mL was diluted directly by 2 mL distilled water, and 0.5 mL of the resulting suspension was treated by methanol following the same steps described above. HPLC analysis reported by Najlah et al. [14] was performed on an UltiMate 3000 UHPLC (Thermo Fisher Scientific UK, Loughborough, UK) with a Phenomenex Luna C18 4.6 × 150 mm column with a 5 μm particle size (Phenomenex, Torrance, CA, USA). The mobile phase comprised 80% HPLC grade methanol and 20% HPLC grade water. The flow rate was 1 mL/min, whereas UV detection was performed at a wavelength of 275 nm with an injection volume of 20 μL.

The solubility of DS in water is less than 4.5 mg/L [11,26,27]. Therefore, the amount of the drug dissolved in water during hydration was ignored. However, any traces solubilized during the process were excluded by following the same dilution factor for both the entrapped and total drug calculations. The entrapment efficiency (EE) of DS in liposomes was calculated using the following equation:EE (%) = (Amount of DS entrapped/Total amount of DS in liposomal suspension) × 100

The drug loading efficacies (DLE%) were calculated by the following equation [28]:DLE (%) = (Amount of DS entrapped/Theoretical DSF content in liposome) × 100%

#### 2.2.5. Stability of Encapsulated Disulfiram in Horse Serum

A sample (0.5 mL) of liposome formulation (10 mol% formulation) was preheated at 37 °C, added to 2 mL of horse serum (preheated at 37 °C) and incubated in a shaking water bath at 37 °C (Grant OLS Aqua Pro, Shepreth, UK) and 100 rpm. For free DS, 25 μL of 4 mg/mL DS in DMSO was pipetted in to 2 mL of horse serum diluted with 475 μL of distilled water (preheated at 37 °C). At specific time intervals, aliquots of 200 μL were added to 500 μL of ethanol and vortexed for 1 min. The mixture solution was centrifuged at 10,000× *g* for 10 min (Heraeus Fresco 17, UK). The supernatant “A” was collected and the pellet was re-suspended in 0.5 methanol vortex for 30 s and heated in the water bath 37 °C for 2 min, then vortexed for 30 s again and centrifuged at 10,000× *g* for 10 min. The supernatant “B” was collected, added to supernatant “A” and analyzed by HPLC using the methods reported above. For the control, the same formulation and previous steps were followed, but by replacing the horse serum with distilled water. The control was used as 100% for the stability calculations.

#### 2.2.6. Cytotoxicity Study (MTT Assay)

The H630_WT_ (passage 13–27) and H630_R10_ (passage 4–16) cells were seeded in 96-well plates at seeding density of 1 × 10^4^ cells/well in Dulbecco’s modified Eagle’s medium (DMEM) with 10% FBS, 1 mM sodium pyruvate, 2 mM *L*-glutamine and 0.1 mM non-essential amino acids. Cells were incubated at 37 °C, 5% CO_2_ and 95% relative humidity. Cells were constantly exposed to different concentrations of the 10% *w*/*w* DS-loaded liposomes in combination with 10 μM of CuCl_2_ for 72 h and then subjected to a standard 3-(4,5-Dimethylthiazol-2-yl)-2,5-Diphenyltetrazolium bromide (MTT) assay as previously described [29]. Free DS has been used as a positive control. The experiments were carried out in triplicates and the IC_50_ values were calculated. For combination studies, cells were exposed simultaneously to similar concentration range of DS and paclitaxel (1–250 nM) and DS (1–250 nM) and 5FU (1–250 μM).

#### 2.2.7. Statistical Analysis

Statistical significance was measured using the one-way analysis of variance (ANOVA) and student’s t-tests, as appropriate. All values were expressed as the mean ± standard deviation. Values with *p* < 0.05 were regarded as significantly different.

## 3. Results and Discussion

Several DS liposomal formulations were prepared using three phospholipids HSPC, DDPC or DSPE-PEG_2000_, with cholesterol (Ch) in 50:50 lipid:Ch molar ratios (Table 1). We have reported in a previous study that liposomes prepared at an equimolar ratio of lipid:Ch using ethanol-based proliposome technology were able to act as potential carrier of the highly hydrophobic anticancer drugs [20]. DS was included in the liposomal formations at a molar percentage ranging from 0 to 15 (mol%) of the lipids (Table 1).

### 3.1. Size Analysis of Liposomes

In recent decades, a few techniques have been predominantly employed to reduce the size of liposomes into the nano range, such as probe sonication [30], membrane extrusion [31] and high-pressure homogenization [32]. We have previously reported that high-pressure homogenization (HPH) might be more advantageous than probe sonication; HPH produced vesicles with a similar drug entrapment efficiency and superior homogenization output rate [23]. Furthermore, HPH overcame problems associated with probe sonication such as sample contamination (with titanium particles leached from the probe) and overheating [23]. In this study, HPH was used to reduce the size of liposomes and transform large multilamellar vesicles (MLVs) into smaller unilamellar vesicles (SUVs, ~100 nm).

As shown in Figure 2a, HPH was successfully employed to produce liposomes with a range of sizes around 100 nm. For all formulations, the drug inclusion had no significant impact on liposome size (*p* > 0.05). Similarly, the size of produced vesicles was not affected by lipid phase composition as there were no significant difference (*p* > 0.05) amongst HSPC formulations and their corresponding DPPC formulations. Noteworthy, the average size of non-PEGylated formulations (both HSPC and DPPC) was about 100 nm, while the average size of PEGylated formulations was around 80 nm. This might be considered as a general trend that similar size reduction methods can produce PEGylated liposomes that are smaller than their non-PEGylated counterparts. This trend has been confirmed in the literature and explained by the increased intensity of lateral repulsion caused by the addition of PEG to the lipid bilayers (i.e., the lipid bilayer will curve; this will reduce the size of vesicle). Furthermore, PEGylation decreases lamellarity as a result of increased interlamellar repulsion [33]. Nonetheless, in this study, no significant difference was proven by the statistics (Figure 2a).

It has been reported that the size of liposomes might be influenced by the degree of phospholipid saturations, and the charge and/or the length of the chain of their lipid components [20,34,35]. In this study, the only difference between both phospholipids is the chain length (16 and 18 for DPPC and HSPC, respectively); hence, no significant changes in the measured size were shown. Differently, we have reported previously that paclitaxel (PTX)-loaded liposomes made from HSPC had larger sizes than that made from DPPC [20]. This might be due to different strength in drug–lipid-bilayer interactions between both sets of formulations.

HPH has also generated liposomes with relatively narrow size distribution (i.e., low PI), regardless of lipid type, PEGylation and drug concentration (Figure 2b). The polydispersity index (PI) for all liposomes was found to be below 0.35 (Figure 2b). However, PEGylation was found to produce liposomes of narrower PIs (*p* < 0.05) than that of the conventional PEGylated liposomes, regardless of lipid type and drug concentration (Figure 2b). This is attributed to the superiority of PEGylated liposomes over conventional vesicles [36,37]. For non-PEGylated liposomes, the inclusion of DS resulted in liposomes with narrower PIs than that of empty liposomes. Surprisingly, no similar trend was observed for the PEGylated liposomes. This might be due to a less effective drug–bilayer interaction compared to the effect of PEGylation, leading to better curving of the bilayer, as explained above. It is noteworthy to mention that PI was independent of drug concentration for all loaded formulations (Figure 2b).

During the development stage of this method, our results showed that PI and size measurements were dependent on the number of homogenization cycles up to a certain number (data not shown). Conclusively, the low PI for all formulations indicates that the number of cycles selected was appropriate and no further high-pressure homogenization was required.

### 3.2. Zeta Potential Analysis

The zeta potential (ZP), the electrostatic charge of the particle surface, plays an important role in controlling the stability of colloids by the repulsive energy barrier opposing the aggregation of dispersed particles (e.g., liposomes) in buffer solutions [37,38]. The ZPs of all liposomes were negative, irrespective of formulation. Furthermore, for all formulations, insignificant effects (*p* > 0.05) of drug concentration on the zeta potential were observed (Figure 3). This may indicate that HPH produced stable nano-liposomes vesicles, as the electrostatic repulsion between negatively charged vesicles may reduce aggregation [38].

PEGylation of DS-free liposomes resulted in a significant increase (*p* < 0.05) in the negativity of zeta potential for both phospholipids. However, despite the apparent differences in ZP values between PEGylated and non-PEGylated DS-liposomes, no statistically significant differences amongst DS-loaded liposomes were found. Nevertheless, this might suggest that ZP values might be affected by vesicle size [38].

### 3.3. Drug Loading and Entrapment Efficiencies of DS in Liposomes

In a liposomal dispersion, drug is expected to be entrapped in the liposomal vesicles (in the core for hydrophilic drugs or embedded within the lipid bilayer for hydrophobic drugs). Additionally, this process depends on many factors such as the ion strength and the pH of the aqueous phase, the incubation time, the drug to lipid loading ratio, the lipid composition and other [39]. However, the excess amount normally remains as crystals suspended, aggregated (within the aqueous phase), or sedimented [40]. During HPH, samples are forced under high pressure through a narrow nozzle orifice followed by several narrow gaps created by the reactors. Samples are then collected and returned to the feeding chamber to start the next cycle. During this process, unentrapped drug particles may be filtered out of the liposomal formulations and trapped between the reactors and in the tubing. Therefore, it is very important to develop methods that can capture both the amount entrapped by liposomes and the actual total amount in the liposomal formulation (the entrapped plus the free drug).

To present an accurate description of loading amounts of DS to liposomes, two efficiencies were calculated. Namely, (1) Drug Loading Efficiency (DLE%) to calculate the amount of drug loaded into liposomes out of the theoretical total drug amount used in each formulation; and (2) entrapment efficiency (EE%) to calculate the amount of DS entrapped in liposomes out of the actual total amount in each liposome formulation after size reduction.

As shown by Figure 4a, DLE% of DS decreased with increasing drug/lipid ratio (*p* < 0.05) and that was independent of phospholipid composition. For example, the DLE% of DS in HSPC-liposomes was 91.7% ± 7.1% for the 5 mol%, and that was decreased to 48.12% ± 12.0% for 15 mol% (*p* < 0.05). However, DLE% for all liposomes was generally similar for the same drug/lipid ratio over the formulations regardless phospholipid composition or PEGylation (Figure 3). Similar findings were observed for paclitaxel (PTX)-liposomal formulations, processed by HPH (unpublished data). Whereas, in our previous studies, using probe sonication for size reduction, lipid composition had a significant effect on PTX entrapment efficiency [20]. This suggests that lipid composition and/or PEGylation had a minimal influence on drug loading for liposomes that have undergone size reduction by HPH.

Figure 4a also shows that DS loading reached a plateau phase in all formulations at approximately 7 mol%. More explicitly, for the 10 mol% formulations, DLE% values were around 60%–68%, which correlates with 6–6.8 mol% actual entrapment of DS. Similar calculations for 15 mol% will lead to a 7 mol% actual entrapment of DS. Therefore, no significant increase in DS loading was observed by increasing the initial loading ratio of DS. These results became more obvious when we observed that the EE% for all formulations were similar for both 15 mol% and 10 mol% formulations as shown in Figure 4b (no significant differences; *p* > 0.05). This means that liposome bilayers were not able to entrap greater DS proportions, i.e., a maximum interaction between DS and the phospholipid was reached.

Figure 4b presents the entrapment efficiency of DS liposome formulations. Despite the decrease in EE% for the 10 and 15 mol% compared to that of 5 mol%, the entrapment efficiencies were above 80%. This indicates that more than 80% of DS are entrapped in the liposomal vesicles of the final formulations. In other words, less than 20% of the DS is free in the final formulation. This might be considered—to an extent—a benefit for HPH as this size reduction technique was able to remove most of the free drug from the final formulation. However, this also might initiate challenges during the scale-up process, such as cleaning the kit after each preparation and validating methods to avoid cross contamination.

We have previously studied the influence of lipid composition (such as HSPC:Ch and DPPC:Ch in equimolar ratios) on the physicochemical properties of formulations and the entrapment efficiency of paclitaxel (PTX) in the liposomes [20]. In that study, the entrapment efficiency of PTX was found to be dependent on lipid composition with DPPC showing the highest loading efficiency (4.2 mol% at the maximum concentration). Unlike PTX liposomes, DS entrapment showed significantly higher mol%, almost a double of that found with PTX and without significant difference between HSPC and DPPC. These differences might be due to the difference in the physiochemical properties between DS and PTX, although both are lipophilic drugs. A main difference between DS and PTX is that the molecular weight is two-times more for PTX than that of DS; this might explain the higher mol% entrapment for DS.

It appears that the molecular weight of the loaded drug plays an important role in controlling loading efficiency. For large molecules such as PTX, the maximum loading concertation was found to be 3–4.2 mol% [15,20]. However, maximum loading concentration as high as 11.2 mol% was reported for smaller molecules such as brucine [41]. Many other factors may also influence maximum loading, such as the electrostatic interaction between the drug and polar head groups, which is beyond the scope of the current study and more investigations are needed in the future.

### 3.4. Stability Studies in Horse Serum

Although the anticancer activity of DS has been known for many years, its application in cancer therapy is limited by its instant degradation in the bloodstream (t_1/2_ < 4 min) [8]. The orally administrated DS undergoes extensive first-pass metabolism in the liver; hence, the oral administration of DS is not suitable for cancer treatment. This may explain the disappointing results from several oral DS-based cancer clinical trials according to ClinicalTrials.gov (http://www.clinicaltrials.gov/).

Liposomes are already established as anticancer drug delivery systems with evidence of capability to overcome the undesirable physicochemical properties of many anticancer drugs. Liposome-loaded DS formulations can be applied intravenously to deliver the drug at cancer cells; this can avoid liver-enrichment and protect the essential thiol groups on their way to the cancer site. In this study, DS was encapsulated into liposomes and PEGylation was used to further extend the half-life of DS and provide further prospective long circulation in the blood. The stability of DS was studied in horse serum to investigate the efficiency of resulting PEGylated liposomes to protect DS.

As shown in Figure 5, both HSPC and DPPC liposomes demonstrated similar stability profiles, with significantly (*p* < 0.05) enhanced stability of loaded DS compared to that of the free drug. Almost 70% of free DS was degraded within the first 20 min, whereas an equal amount of loaded DS at both HSPC and DPPC liposomes took more than one hour to disappear. Furthermore, this time was extended to more than three hours upon PEGylation, regardless of the phospholipid composition. These differences are confirmed by the half-lives of DS in horse serum, which are extended by approximately three-fold for DS-loaded HSPC (25.8 ± 6.5 min) and DPPC (28.7 ± 4.5 min) compared to that of free DS (9.7 ± 2.3 min). The longest half-lives of DS were obtained after PEGylation; 71.9 ± 12.3 min for PEG-HSPC and 77.3 ± 9.6 min for PEG-DPPC (Figure 5b). Although the results show a slight difference between HSPC and DPPC liposomes, no statistically significant difference (*p* > 0.05) was observed between both phospholipids.

PEGylation has been used in many studies to enhance the stability and prolong the circulation of liposomes in vivo. The mechanisms protective action of PEG has been extensively investigated. One of the most popular opinions is that the interactions of blood components with the liposome surface are sterically hindered by the PEG coating the liposomes. This averts opsonization and uptake by the mononuclear phagocyte system (MPS) [42,43]. Furthermore, surface modification of liposomes by a soluble, well hydrated and chemically inert polymer (PEG), will decrease the surface hydrophobicity and increase the repulsive interactions between colloidal particles (i.e., less liposome aggregation and reduced binding of plasma proteins) [42,44].

PEG liposomal formulations have been successfully used to resolve drug delivery problems of many drugs. For example, the first FDA-approved PEGylated liposomal formulation, Doxil^®^, exhibited prolonged retention in vivo and improved therapeutic effects [45]. However, for some drugs, such as vincristine, the result was unsatisfactory because its rapid leakage compromised its prolonged circulation [43]. Therefore, using PEGylated liposomes should be subjected to in depth investigation, on a case by case basis, depending on the drug and proposed application.

In this study, we found that PEGylated liposomes have a great potential for improving the stability of DS and provide a prospective prolonged circulation in the blood. These results are also supported by a recent study using DSPE-PEG_2000_-based PEGylated copper oleate liposome (Cu(OI)2-L) for targeted delivery in cancer. Pharmacokinetic studies demonstrated that Cu(OI)2-L had a prolonged circulation time compared to that of injected copper oleate solution [46]. However, no stability study results in serum were shown and the feasibility (and compatibility) of combining DS with Cu^+2^ in the same dosage form should be investigated in depth. For our study, more investigation is needed to explore the long-term stability of the liposomal formulations and their ability to protect DS in vivo. This has been already listed on to our future plans.

### 3.5. Cytotoxicity Studies

When repetitively exposed to chemotherapeutics, cancer cells become more resistance and aggressive. This is considered as one of the major hurdles to successful chemotherapy. In this study, 5-fluorouracil (5FU)-resistant colon cancer cell line H630_R10_ and their parental wide-type (sensitive) cell line H630_WT_ were used to determine the cytotoxicity of liposomes loaded with a range of DS concentrations using the MTT assay. The resistant cancer cells are commonly cross-resistant to a wide range of anticancer drugs, Figure 6 shows that H630_R10_ cell lines were not only resistant to 5FU, but also resistant to PTX at concentrations higher than 1000 nM, whereas the sensitive H630_WT_ were completely killed at concentration far below 1000 nM. These results are confirmed by IC_50_ being above 1000 nM and 43.63 ± 15.21 for PTX (*p* < 0.01) in H630_R10_ and H630_WT_ cell lines, respectively (Table 2). The microscopy images taken 72 h after treatment also confirm that H630_WT_ cell line, showing apoptotic signs, were affected by both drugs 5FU and PTX. On the other hand, H630_R10_ cell lines were not affected by both 5FU and PTX at the same corresponding concentrations and showed no features of apoptosis (Figure 7).

Figure 8 shows the effect of DS liposome formulations on the viability of H630_R10_ and H630_WT_ cells. Cell viabilities of both cell lines were dependent on drug concentrations regardless phospholipid type (*p* < 0.05). The empty liposomes (no DS), used as a negative control, had no effect on the viability of both cell lines (Appendix A). This may suggest that the remaining ethanol resulting from the preparation methods had no contribution to the cytotoxicity of DS-loaded formulations. For both cell lines, DS completely inhibited the cell proliferation at the highest concentration (250 nM). Despite the apparent trend showing that the cytotoxicity of DS alone on both cell lines was also higher than that entrapped in liposome formulations, no statistically significant difference was found between the free and encapsulated drug (Figure 8, Table 2). Similarly, IC_50_ values of DS formulations show that the DS and its liposomal formulations were more toxic toward H630_R10_ than the sensitive cells H630_WT_; these differences are statistically insignificant (*p* > 0.05). The apparent reduced cytotoxicity of DS in liposome formulations might be attributed to the slow release profile of the drug from the formulations (Appendix A), or to nutritious lipids forming liposomes. It is worth noting that PEGylation had no significant effect (*p* > 0.05) on the cytotoxicity profiles of HSPC or DPPC DS-loaded liposomes despite the apparent differences amongst IC_50_ values for PEGylated and non-PEGylated liposomes. More investigation is necessary to study the influence of PEGylation on the cellular uptake of DS-loaded liposomes. However, we believe that the cytotoxicity of DS-loaded liposomes is more dependent on the DS release form the liposome rather than liposomal interaction with the cellular membrane.

The cytotoxic effect of DS liposomal formulations was confirmed by microscopy images taken after 72 h of exposure to the tested formulations (Figure 7). Both the H630_WT_ and H630_R10_ cells demonstrated apoptotic morphologies (cell blebbing and nuclear condensation and fragmentation) for DS and its PEGylated liposomoal formulations. Similarly, as mentioned earlier, apoptotic morphology was shown by H630_WT_ after exposure to PTX. In contrast, PTX had no effect on H630_R10_ due to pan chemoresistance. To assess any potential synergic effect between DS/Cu and 5FU or PTX, a combination cytotoxicity has been performed using MTT assay. The resistant H630_R10_ cell lines were exposed to serial concentrations of 5FU or PTX and DS/Cu or its PEGylated formulation PEG:HSPC:Ch. As shown in Figure 9, the combination between DS/Cu and 5FU or PTX resulted in enhanced cytotoxicity which was significantly (*p* < 0.05) higher than the cytotoxic effect observed with each of the drugs applied individually. More importantly, DS/Cu and its PEGylated liposomal formulation were able to reverse the chemoresistance of H630_R10_. The IC_50_ values of the combinations were significantly (*p* > 0.05) reduced compared to that of sole exposure to each drug. For example, the IC_50_ value for PEG:HSPC:Ch combined with PTX was 21.81 ± 4.15 compared to that PTX alone (over 1000 nm).

Drug-free formulations had no toxic effect on both colorectal cell lines. This has been confirmed previously by several studies; drug-free liposomes displayed a neutral effect on both cancer and normal cell lines in vitro [47]. This may be expected as liposomes are, in general, made of natural biocompatible molecules similar to those forming cellular membranes. Furthermore, liposome vesicles might enhance the efficacy of the drugs by binding to the cells and releasing the therapeutic molecules in a sustained manner [47,48].

In conclusion, in this paper, we have reported the development of PEGylated liposomes of disulfiram using ethanol-based proliposome methods followed by high-pressure homogenisation for size reduction. DS-loaded PEGylated liposomes had relatively smaller sizes and significantly lower polydispersity index than those of conventional liposomes. All liposomal formulations had good entrapment efficiencies and maintained the cytotoxic effect of disulfiram in the presence of copper. More importantly, we found that PEGylated liposomes significantly improved the stability of DS in horse serum. Therefore, PEGylated liposomes might be able to act as potential nanocarriers of DS in cancer therapy. The ability of PEGylated liposomes to protect DS and to provide prolonged circulation in vivo has been already listed in our future plans.

## Figures and Tables

**Figure 1 pharmaceutics-11-00610-f001:**
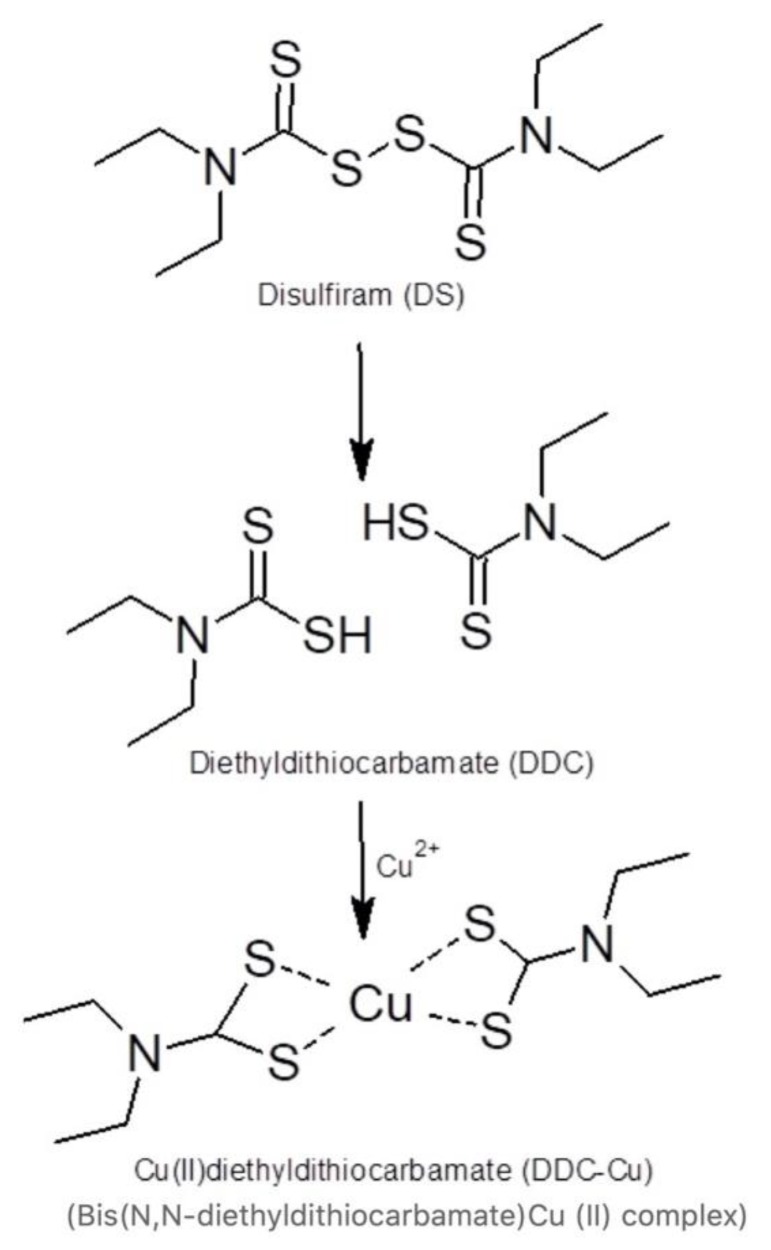
Disulfiram (DS) reaction with copper.

**Figure 2 pharmaceutics-11-00610-f002:**
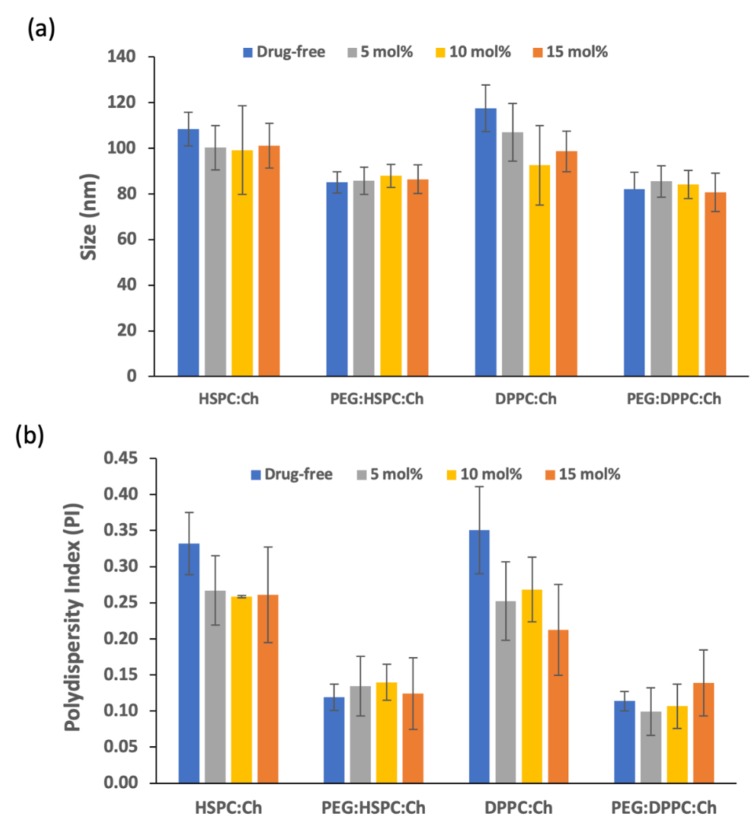
Size (Z_average_) (**a**) and PI (**b**) of liposomes after high-pressure homogenization with a range of DS concentrations (*n* = 3 ± SD).

**Figure 3 pharmaceutics-11-00610-f003:**
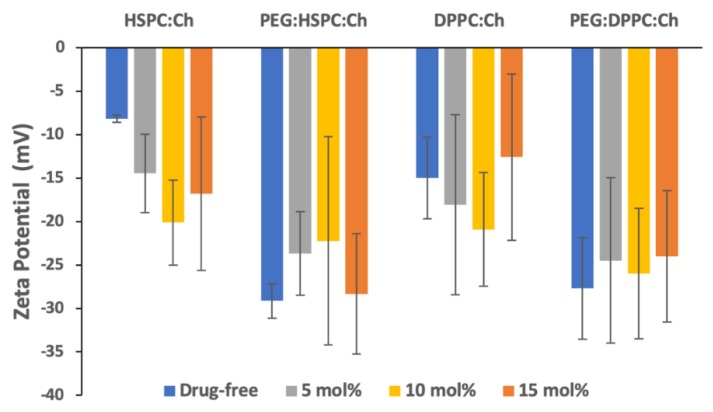
Zeta potential values of liposomes with a range of DS concentrations (*n* = 3 ± SD).

**Figure 4 pharmaceutics-11-00610-f004:**
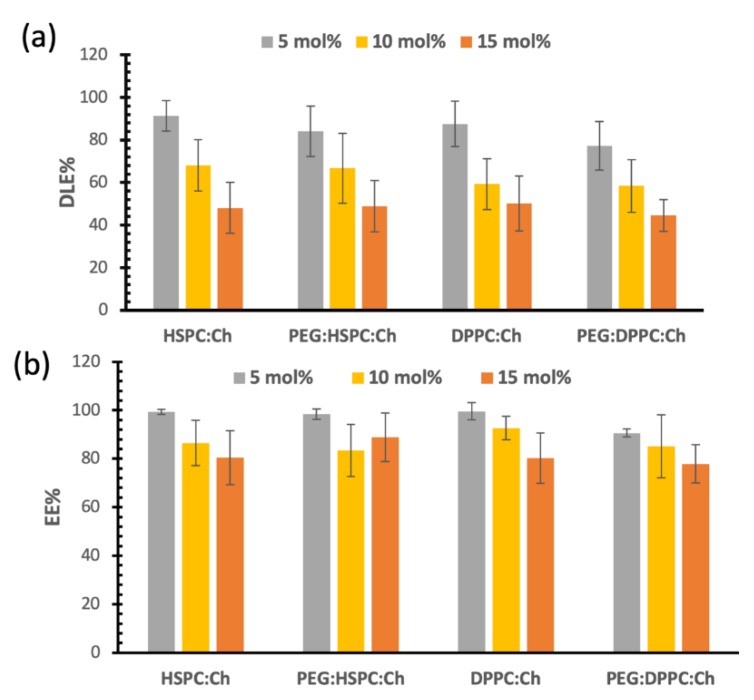
(**a**) Drug loading efficacies (DLE%), (**b**) entrapment efficiency (EE%).

**Figure 5 pharmaceutics-11-00610-f005:**
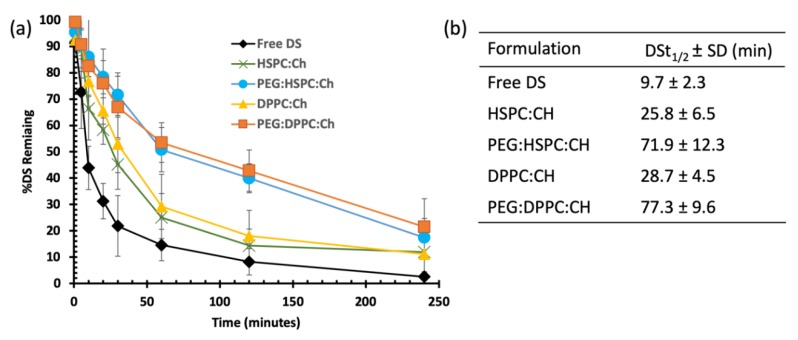
The influence of liposomal formulation on the stability of DS in horse serum: (**a**) degradation curves and (**b**) the half-lives (t_1/2_) of DS in different formulations.

**Figure 6 pharmaceutics-11-00610-f006:**
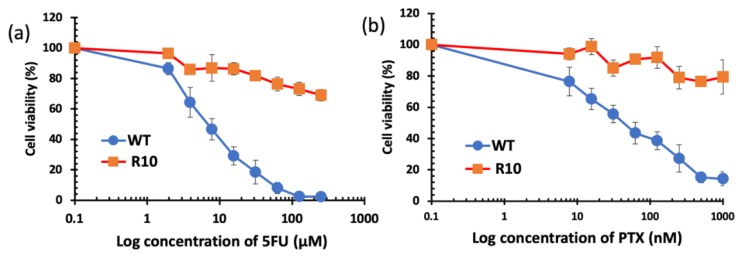
Survival curves (MTT cytotoxicity assay) of colorectal cancer cell lines H630_WT_ and H630_R10_ with increasing concentrations of 5FU (**a**) and PTX (**b**) (*n* = 3 ± SD).

**Figure 7 pharmaceutics-11-00610-f007:**
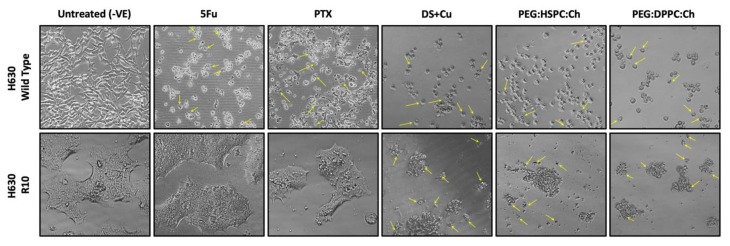
The morphology (×100 magnification) of parental and resistant cells after 72 h of exposure to 5FU (100 µM), PTX (125 nM), and DS formulations (equivalent to 125 nm of DS). The arrows are pointing to features of apoptosis.

**Figure 8 pharmaceutics-11-00610-f008:**
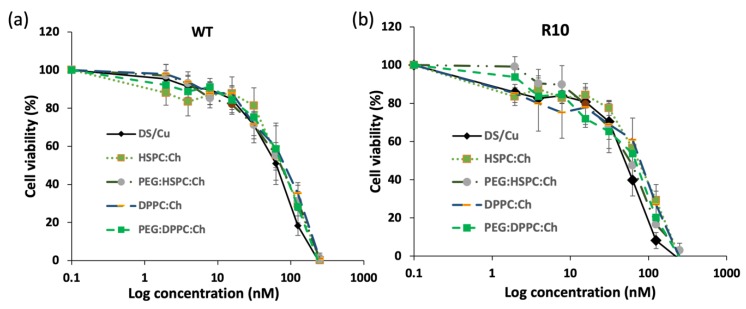
Survival curves of the MTT cytotoxicity assay for DS formulations on colorectal cancer cell lines (**a**) H630_WT_ and (**b**) H630_R10_ (*n* = 3 ± SD).

**Figure 9 pharmaceutics-11-00610-f009:**
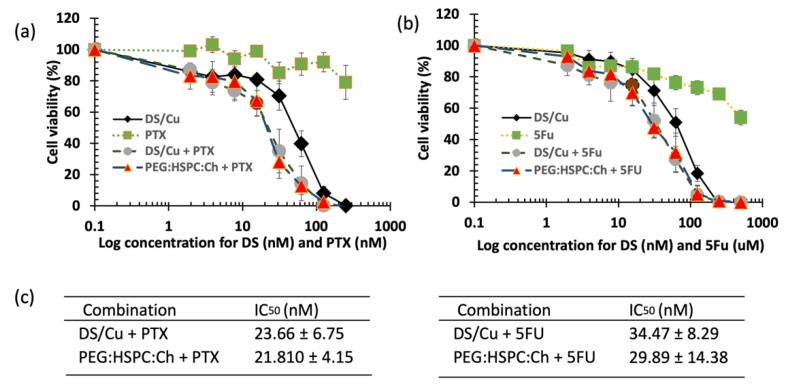
The MTT cytotoxicity assay. H630 R10 cell lines were exposed to different concentrations of DS formulations in combination with (**a**) PTX (**b**) 5FU. (**c**) The IC_50_ values of DS formulations in combination with (**a**) PTX (**b**) 5FU (*n* = 3 ± SD).

**Table 1 pharmaceutics-11-00610-t001:** The ingredients of DS liposomal formulations.

	Ingredients	DSPE-PEG2000	HSPC	DPPC	Ch	DS
Formulation						
HSPC:Ch	0.0 (mol% *)	-	1 **	-	1	-
	5.0	-	1	-	1	0.11
	10.0	-	1	-	1	0.22
	15.0	-	1	-	1	0.36
PEG:HSPC:Ch	0.0	0.1	0.9	-	1	-
	5.0	0.1	0.9	-	1	0.11
	10.0	0.1	0.9	-	1	0.22
	15.0	0.1	0.9	-	1	0.36
DPPC:Ch	0.0	-	-	1	1	-
	5.0	-	-	1	1	0.11
	10.0	-	-	1	1	0.22
	15.0	-	-	1	1	0.36
PEG:DPPC:Ch	0.0	0.1	-	0.9	1	-
	5.0	0.1	-	0.9	1	0.11
	10.0	0.1	-	0.9	1	0.22
	15.0	0.1	-	0.9	1	0.36

* Drug lipid molar percentage. ** Molar ratio.

**Table 2 pharmaceutics-11-00610-t002:** The IC_50%_ values of DS formulations on both H630_WT_ and H630_R10_ (*n* = 3 ± SD).

IC_50_	H630 WT (nM)	H630 R10 (nM)
5FU	3420 ± 630.0	>250,000
PTX	43.63 ± 15.21	>1000
DS/Cu	57.736 ± 11.08	49.092 ± 8.20
HSPC:Ch	75.544 ± 22.68	73.094 ± 12.28
PEG:HSPC:Ch	69.076 ± 3.95	56.800 ± 3.21
DPPC:Ch	76.273 ± 10.97	64.69 ± 10.88
PEG:DPPC:Ch	71.289 ± 10.81	56.165 ± 21.16

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
