# Peer review of "Development of Injectable PEGylated Liposome Encapsulating Disulfiram for Colorectal Cancer Treatment"

_pharmaceutics, 2019, doi:10.3390/pharmaceutics11110610_

Round 1

Reviewer 1 Report

This research described about therapeutic efficacy and the generation method to produce PEGylated liposome encapsulated with disulfiram (DS) by various lipid phase compositions for the treatment of colorectal cancer. DS liposomal formulations generated by HSPC, DDPC, or DSPE:PEG2000 under various production method and condition were characterized and analyzed. When liposome formulations consisting of various phospholipid types with PEG were generated with various anti-cancer drugs, including DS, cancer cell killing effects increased in drug-resistant colon cancer cells. This paper is well written, combining results and discussions. The experiments are well planned and the technical quality is not doubted. However, several issues need to be addressed in order for the manuscript to be published in Pharmaceutics. The reviewer offers following comments for improving the manuscript.

⦿ Major comments:

1. The size of the liposome and the PI value were reduced by PEGylation (Figure 2). In addition, the negative charge on liposome surface has increased by PEGylation (Figure 3). An increase in surface negative charges of liposome will increase the repulsion force on the surface of cells with negative charges, which will reduce the efficiency of cellular uptake. Also, the use of PEG will induce the decreased transduction efficiency by masking in sites related to cellular uptake of liposome, but why is the IC50 value of PEGylated liposome improved more than the value of non-PEGylated liposome (Table 2) ? In other words, why does it work better in cells?

2. The usage of PEG has many advantages when administered in vivo, such as the reduction of interaction with blood components, leading to the increased blood circulation time and the decreased metabolic action of DS in the liver etc.. Simple animal experiment (blood clearance or liver toxicity) needs to be shown in this study for therapeutic efficacy and superiority of DS-loaded liposome generated by your system, although you mentioned as your next plan.

⦿ Minor comments:

1. In Figure 5, please modify the terms for free DSF or free DS as one.

2. In Figure 8, there are too many groups to verify the results properly. There is also no description of (E). Please subtract the result of Empty liposome into the supplementary data or divide the graph.

3. You described the characteristics of apoptosis of the cells treated with DS liposome or other anti-cancer drugs (page 13, line 445). Please mark those morphological features (cell blebbing and nuclear condensation etc.) with arrows.

4. Please unify the description of the X-axis of the graph in Figure 9. For example, “Log concentration for DS (nM) and PTX (nM)” in Figure 9A.

5. Please write a coherent expression of words in Manuscript, like h or hour, RT or room temperature, or italic font of in vivo etc..

6. Please recheck present and past representations in describing the results of experiments.

Author Response

We thank the reviewers for their review and valued comments, we have responded to every comment made and have amended the manuscript in order to make the suggested improvements. We believe that the amended manuscript is now of higher quality in terms of presentation and meets the high standards of Pharmaceutics.

Our detailed responses to all of the reviewer’s comments are shown below.

Response to comments from Reviewer 1

Reviewer 1: “This research described about therapeutic efficacy and the generation method to produce PEGylated liposome encapsulated with disulfiram (DS) by various lipid phase compositions for the treatment of colorectal cancer. DS liposomal formulations generated by HSPC, DDPC, or DSPE:PEG2000 under various production method and condition were characterized and analyzed. When liposome formulations consisting of various phospholipid types with PEG were generated with various anti-cancer drugs, including DS, cancer cell killing effects increased in drug-resistant colon cancer cells. This paper is well written, combining results and discussions. The experiments are well planned and the technical quality is not doubted. However, several issues need to be addressed in order for the manuscript to be published in Pharmaceutics. The reviewer offers following comments for improving the manuscript.”

We thank the reviewer for the recommendation and the constructive feedback. We have addressed all points as shown below.

⦿ Major comments:

The size of the liposome and the PI value were reduced by PEGylation (Figure 2). In addition, the negative charge on liposome surface has increased by PEGylation (Figure 3). An increase in surface negative charges of liposome will increase the repulsion force on the surface of cells with negative charges, which will reduce the efficiency of cellular uptake. Also, the use of PEG will induce the decreased transduction efficiency by masking in sites related to cellular uptake of liposome, but why is the IC50 value of PEGylated liposome improved more than the value of non-PEGylated liposome (Table 2)? In other words, why does it work better in cells?

We agree with the reviewer, the PEGylated liposomes may appear to be more toxic than conventional liposomes. However, there was no evidence by the statistics that this difference was significant. We believe that the slow release might allow slight differences in cytotoxicity profiles over 72 hours of incubation (Figure 1S). However, more investigation is needed to study the influence of PEGylation on the cellular uptake of DS loaded liposomes. As a response to this comment, we have added a small paragraph to clarify this point (lines 461-466).

The usage of PEG has many advantages when administered in vivo, such as the reduction of interaction with blood components, leading to the increased blood circulation time and the decreased metabolic action of DS in the liver etc.. Simple animal experiment (blood clearance or liver toxicity) needs to be shown in this study for therapeutic efficacy and superiority of DS- loaded liposome generated by your system, although you mentioned as your next plan,

We also agree with the reviewer that in vivo study is the next step to assess the efficiency of this liposomal formulations, we clearly stated that PEGylation must be considered for each drug as a separate case. At this stage, this manuscript shows that PEGylated liposomes were able to provide good protection to DS in horse serum, hence they might have great potential as carriers for DS. We hope that animal facilities will be available in the near future to profoundly explore this potential.

⦿ Minor comments:

We thank the reviewer for these detailed and very useful comments that have improved the quality of this manuscript.

In Figure 5, please modify the terms for free DSF or free DS as one.

Done

In Figure 8, there are too many groups to verify the results properly. There is also no description of (E). Please subtract the result of Empty liposome into the supplementary data or divide the graph.

The results of empty liposomes have subtracted into the supplementary data (Figure S3) and a description of (E) has been added. Figure 8 has been amended.

You described the characteristics of apoptosis of the cells treated with DS liposomes or other anti-cancer drugs (page 13, line 445). Please mark those morphological features (cell blebbing and nuclear condensation etc.) with arrows.

Done – Figure 7

Please unify the description of the X-axis of the graph in Figure 9. For example, “Log concentration for DS (nM) and PTX (nM)” in Figure 9A.
Please write a coherent expression of words in Manuscript, like h or hour, RT or room temperature, or italic font of in vivo etc..

Done- please see track changes.

Please recheck present and past representations in describing the results of experiments.

Done- please see track changes.

Reviewer 2 Report

In the present study, the authors developed PEGylated liposome encapsulating dislfiram (DS).

However, additional discussion or experiment would be needed before the publication. Please read specific comments described below.

The authors confirmed the stability of DS encapsulated in PEGylated liposome in plasma (Figure 5), but there is no information on release rate of DS from PEGylated liposome in plasma, which is related with pharmacokinetics and pharmacodynamics of DS encapsulated in PEGylated liposome in vivo. The authors mentioned that the aim of the present study is to develop long circulating DS-loaded PEGylated liposomes in the Introduction, but there is no data on plasma retention after intravenous injection in vivo. The authors should prove it to achieve the goal.

Author Response

We thank the reviewers for their review and valued comments, we have responded to every comment made and have amended the manuscript in order to make the suggested improvements. We believe that the amended manuscript is now of higher quality in terms of presentation and meets the high standards of Pharmaceutics.

Our detailed responses to all of the reviewer’s comments are shown below.

Response to comments from Reviewer 2

"In the present study, the authors developed PEGylated liposome encapsulating dislfiram (DS). However, additional discussion or experiment would be needed before the publication. Please read specific comments described below.

 The authors confirmed the stability of DS encapsulated in PEGylated liposome in plasma (Figure 5), but there is no information on release rate of DS from PEGylated liposome in plasma, which is related with pharmacokinetics and pharmacodynamics of DS encapsulated in PEGylated liposome in vivo. The authors mentioned that the aim of the present study is to develop long circulating DS-loaded PEGylated liposomes in the Introduction, but there is no data on plasma retention after intravenous injection in vivo. The authors should prove it to achieve the goal."

We thank the reviewer for their comments. The release studies were actually performed and submitted with supplementary data in the first round of submission (Figure S1). The PEGylated liposomes showed a sustained release of DS. This has been mentioned in the manuscript (P12, line 460 and 465). The aim of this study is to develop PEGylated liposomes to enhance the bio-stability of DS, this was proven by the results of stability study in horse serum. Despite the fact that PEGylation has been used widely to ensure long circulations, we have removed “long-circulating” form the introduction (p3, line 98) to avoid any confusion.  Unfortunately, it is impossible to perform in vivo studies at this stage, but it has been already listed on the top of our future plans.

Round 2

Reviewer 1 Report

The authors responded well to all comments.

Overall, the experiment is well designed, and the results are well described logically.

However, although it is somewhat regrettable that there is no in vivo experiment on this DS-loaded liposome, I will look forward to the authors' next paper.

Manuscript is ready for publication.

Thank you.

Reviewer 2 Report

The revised manuscript has been improved. I think that this manuscript is acceptable for this journal.